# INTERACTIVE-ACTION IMAGE GENERATION VIA SYNTHETIC PHYSICAL PRIORS

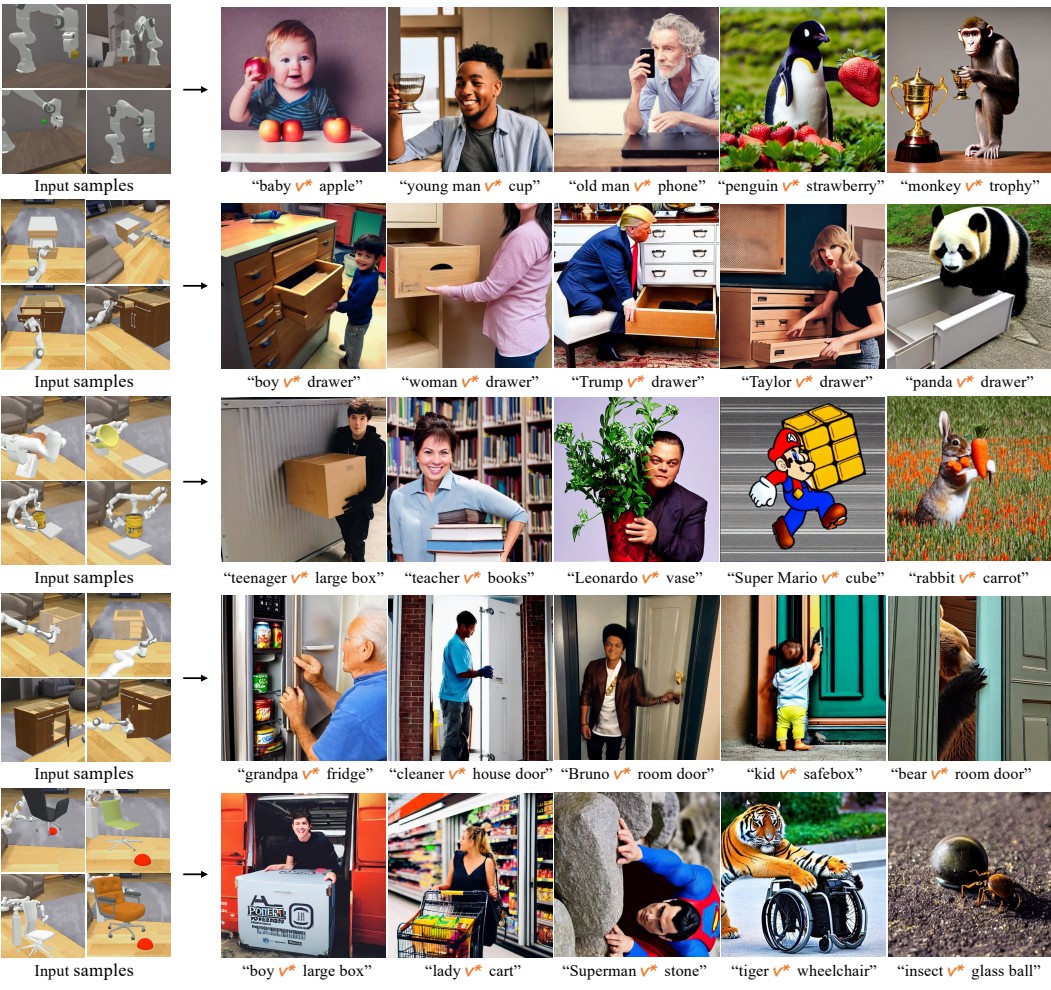

## ABSTRACT

While diffusion-based text-to-image generation has made notable advancements, generating accurate images containing interactive actions remains a challenge due to the lack of inherent physical and spatial priors. To address this problem, we propose a novel pipeline that synthesizes a dataset enriched with physical priors using a graphics engine, combined with a captioning technique. Building on the dataset, we introduce a distillation-structured fine-tuning method, where a teacher network assists in inverting the semantics of interactive actions, leveraging the synthesized priors effectively. This fine-tuning method disentangles the synthetic data features while mitigating random misalignment during the fine-tuning process. Extensive experiments demonstrate that our method not only achieves state-of-the-art results but also highlights the synthetic data's potential to be applied more broadly in enhancing the generation of interactive action images.

# 1 INTRODUCTION

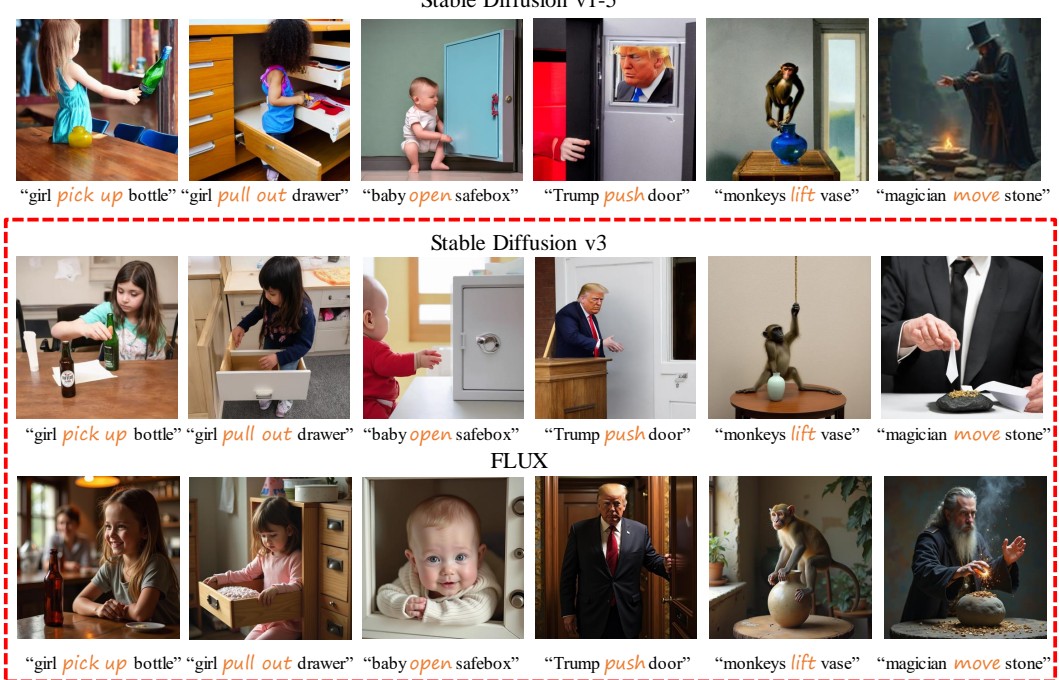

Figure 1: **(for Reviewer WdtC' W1, Esr5' Q4)** Examplar results of Stable Diffusion v1-5, Stable Diffusion v3 Sauer et al. (2024) and FLUX. These results show that existing T2I model can draw the subjects and objects precisely, according to the input prompt, but cannot well generate the interactive action: subjects in these images have unreasonable spatial relationship or do not touch the objects.

Recent years have witnessed significant breakthroughs in the field of text-to-image generation, particularly with powerfulness of diffusion models Gal et al. (2022). However, existing text-to-image generation models such as Stable Diffusion (SD) Rombach et al. (2022) struggle to generate images containing interactive actions. As shown in Figure 1: the generated images from SD totally fail to model correct spatial positions of objects and the interactions between them.

In general, generating interactions requires modeling spatial relationships, which is inherently more difficult than generating isolated objects. For instance, depicting an action as seemingly straightforward as "picking up" involves intricate spatial and physical constraints, including the placement of objects, the elevation at which the pick-up occurs, and the degree of arm curvature, etc.

Existing diffusion models lack explicit physical understanding. Typically, text-to-image models Dhariwal & Nichol (2021); Ho & Salimans (2022); Lugmayr et al. (2022); Esser et al. (2021) are pretrained using image-text pairs. This approach has limitations particularly in learning physical constraints, because images involving interactions are very rare in the training dataset, resulting in models that do not thoroughly comprehend spatial dynamics (for example, illustrations where various models fail to depict spatial relations accurately).

To resolve this problem, we propose to fine-tune the diffusion models on synthetic images with interactions rendered from physical simulation graphics engines Savva et al. (2019); Xia et al. (2018; 2020); Savva et al. (2017); Brockman et al. (2016); James et al. (2020); Xiang et al. (2020). A graphics engine can offer a 3D scene where an object interacts with another object conforming to physical constraints. In this paper, we select SAPIEN Xiang et al. (2020) as the main generation engine due to its convenience, but we also prove that other popular engine like Unity3D is effective as well. In SAPIEN, we first build a 3D scene where a robot agent is interacting with objects, and then render multi-view 2D image dataset which contains physical priors for diffusion models.

However, using synthetic data's physical prior to learn interactive action is not a free lunch: there is a big domain gap between synthetic data and real data where diffusion models are pretrained

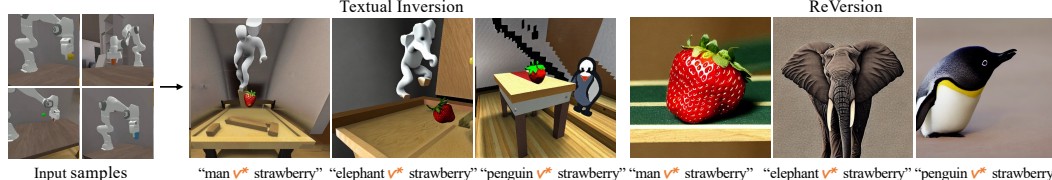

Figure 2: Textual Inversion's problem (left) and ReVersion's problem (right). Textual Inversion tends to overfit to the training synthetic data, maintaining style features of the synthetic data, while ReVersion usually throws the subject or object randomly.

on. We expect to transfer the interactive action learning from synthetic data to real data generation. An intuitive approach is to use textual inversion Gal et al. (2022), a fine-tuning method to learn a specific semantic concept with a rare textual token (in this paper, we use $v*$) in a "Subject-$v*$-Object" prompt, then transfer the concept by using the token for inference. However, optimizing the token's embeddings on synthetic data is prone to overfit, as shown in Figure 2. Our objective is to obtain a generalizable action embedding which disentangles synthetic appearance and style features.

A possible fine-tuning approach is to employ personalization methods like DreamBooth Ruiz et al. (2023). However, DreamBooth requires extra real-images to refine the class-specific prior preservation loss, which helps prevent over-fitting. This requirement cannot be easily meet in our task since real data is very rare. ReVersion Huang et al. (2023b) handles the disentangle problem by analyzing different parts of the textual prompts via a contrastive loss, which steers the embedding towards relation-dense region in the text embedding space. Adapting ReVersion in our task helps to alleviate the overfitting issue but will introduce textual prompt misalignment problem: as shown in Figure 2, subjects or objects will disappear randomly. This problem appears due to the domain gap between the real data and synthetic data.

Therefore we propose a distillation network specified for synthetic data fine-tuning to solve the misalignment problem. We find that although SD fails to accurately model the spatial relation, the subject and object are present without random losing. This observation inspires us to leverage its strengths. As shown in Figure 3: a frozen diffusion model works as a teacher network, which conditions on a "Subject-**and**-Object" prompt, aiming to generate them without considering their relationship; the student network learns the interactive action conditioning on "Subject-$v*$-Object" prompt, disentangling the synthetic features. With a distillation loss, the teacher network offers a guidance signal to supervise the student network to generate the subjects and objects. Then $v*$ can be used to generate images with specific interactive actions as the teaser page shows.

Our contribution is three-folded: 1) we pioneer to focus on interactive-action image generation, an important and challenging task that remains unsolved, and we propose to employ synthetic data to offer more physical priors; 2) we design a new distillation method to fine-tune the diffusion model on synthetic data, solving the misalignment problem introduced by the domain gap; 3) extensive experiments prove the potential ability of synthetic data, our method can generate significantly better images with reasonable spatial relationships and good fidelity.

## 2 RELATED WORK

**Diffusion-based Text-to-Image (T2I) Generation.** With great success, diffusion models have become a mainstream approach in various domains such as image generation Dhariwal & Nichol (2021); Ho & Salimans (2022); Lugmayr et al. (2022); Esser et al. (2021), image processing Saharia et al. (2022b;a); Guo et al. (2023); Chung et al. (2022), video generation Wu et al. (2022); Hong et al. (2022); Wu et al. (2023), and 3D generation Poole et al. (2022); Müller et al. (2023); Nam et al. (2022) etc. Text-to-image (T2I) diffusion models have demonstrated impressive results in converting text descriptions into images. Notable examples include GLIDE Nichol et al. (2021), DALL-E Ramesh et al. (2022), Imagen and Latent Diffusion Model (LDM) Rombach et al. (2022), each contributing unique advancements to the field. Stable Diffusion Rombach et al. (2022), based on LDM, uses a cross-attention mechanism to inject textual conditions into the generation process, significantly improving computational efficiency by operating in latent space. This marks a new era of text-to-image generation capable of handling arbitrarily text descriptions. Motivated by its success and popularity in the open research community, we build our framework on Stable Diffusion.

**Interactive Action Generation.**  A similar task, HOI generation mainly focus on 3D, like DreamHOI Zhu et al. (2024) generate 3D models with deformed SMPL mesh Loper et al. (2023), although the 3D scene be physically constrained, their rendered images have far low fidelity than real images, therefore fine-tuning interactive-action images is more feasible. Due to the demand for generating images with user-specified content, customization methods like DreamBooth Ruiz et al. (2023), Textual Diffusion Gal et al. (2022), Custom Diffusion Kumari et al. (2023), and P+ Voynov et al. (2023) have been proposed. These methods focus on transferring user-specified subject appearance or style by fine-tuning diffusion models on a few images to align a new word with visual semantics, failing to model interactive relationships. ReVersion Huang et al. (2023b) makes progress in learning specific relations denoted by prepositions and modeling simple actions, yet it faces challenges in inverting complex interactive relationships. Meanwhile, recent work ADI Huang et al. (2023a) successfully disentangles actions from exemplar images, but focusing primarily on isolated actions without addressing interactive relationships. Our framework is designed to bridge this gap.

**Interactive Scene Rendering.**  Several simulated environments for robot learning have been developed, each featuring diverse 3D scenes. Habitat Savva et al. (2019)serves as a key example of navigation-focused environments, incorporating elements from Gibson Xia et al. (2018; 2020) and Minos Savva et al. (2017) but it primarily emphasizes static physics over interactive actions. AI2-THOR Kolve et al. (2017) is recognized for its game-like interactive environments, but its navigation is restricted by object interactions. Additionally, platforms like OpenAI Gym Brockman et al. (2016) and RLBench James et al. (2020) provide interactive environments; however, their dependence on commercial software limits their adaptability and customizability. In contrast, the open-source platform SAPIEN Xiang et al. (2020) offers various interactive tasks and part-level physical simulations, providing rich spatial information and priors that can boost our text-to-image generation tasks.

## 3 METHODS

### 3.1 PRELIMINARY

**Stable Diffusion.** Latent diffusion models Rombach et al. (2022) (LDM) refer to a kind of generative models working in latent space, where $\mathcal{E}$ and $\mathcal{D}$ are corresponding encoder and decoder relating pixel space with and latent space, following the mechanism of VQ-VAE Van Den Oord et al. (2017). Specifically, $\mathcal{E}$ encoders an input image $x \in \mathbb{R}^{c \times w \times h}$ into latent space, obtaining corresponding latent code $z_0$ $i.e.$ $z_0 = \mathcal{E}(x)$. Forward diffusion process transforms $z_0$ into $z_T \sim \mathcal{N}(0, I)$ by $T$ steps of gradually adding Gaussian noise $\epsilon \sim \mathcal{N}(0, I)$ with the same shape as $z_0$:

$$q(z_t|z_{t-1}) = \mathcal{N}(z_t; \sqrt{1 - \beta_t}z_{t-1}, \beta_t I), t = 1, ..., T \tag{1}$$

where $q(z_t|z_{t-1})$ is the conditional density of $z_t$ given $z_{t-1}$, $\{\beta\}_{t=1}^T$ are noise scheduler hyperparameters, and $T$ is a large enough parameter. Then the LDM further undertakes the learning of a backward process, aimed at reversing the forward process to reconstruct $z_0$ from $z_T$:

$$p_\theta(x_{t-1}|x_t) = \mathcal{N}(x_t; \mu_\theta(x_t, t), \Sigma_\theta(x_t, t)) \tag{2}$$

After $t = T, ..., 1$ steps backward process, $z_0$ is generated from $z_T$, then the output video results from an decoder, $i.e.$ $x = \mathcal{D}(z_0)$. The backward process is optimized by:

$$\mathcal{L}_{denoise} = \mathbb{E}_{z \sim t, z_0, \epsilon, c} \left\| \epsilon - \epsilon_\theta(z_t, t, c) \right\|_2^2 \tag{3}$$

where $\mathcal{L}_{denoise}$ is a denoising loss. $\epsilon_\theta$ is a conditional denoising network, often implemented by a 2D-UNet. $c$ denotes the condition, if the model is trained with video-text prompt pairs $(x, p)$, $c$ is the conditional text embedding processed by a text encoder $\mathcal{C}$, $i.e.$ $c = \mathcal{C}(p)$.

### 3.2 SYNTHETIC DATA COLLECTION

Synthetic data can be curated based on various platforms. For convenience, we collect the synthetic data on the SAPIEN Xiang et al. (2020), a 3D realistic and physics-rich simulation platform. The platform provides an extensive library of pre-designed articulated robot agents, objects, and habitat assets, enabling users to simulate realistic interactive environments. Specifically, we register an action environment, and then input parameters into the environment to define the objects,

Figure 3: The pipeline of our method. We render synthetic interactive images from a Physical Simulation Environment, which contains rich physical prior, and then utilize them to optimize a placeholder $v*$ to learn an interactive semantic. The denoising loss and steering loss help to disentangle the interactive semantics for $v*$ from the synthetic shallow features, while the teacher network and distillation loss address the misalignment problem introduced by the domain gap.

indoor scenes, agent initial poses, and agent trajectories. Agent trajectories are sampled from various trajectories. Then the robot agent is driven accordingly to finish the interactive task. We input diverse camera viewpoint parameters and render multi-view videos accordingly. Then we sample key-frames in which the robot interacts with objects from the rendered videos to build the image dataset. Details and exemplary generated data can be viewed in Appendix A

Additionally, we provide multiple text templates, which describe the depicted scenes, to pair with each image. These templates are divided into two groups. Specifically, one group is tailored for interactive action modeling and includes a placeholder $v*$, forming "Subject-$v*$-Object" prompts. Another template is the same as the former one, except that $v*$ is replaced by preposition "and", forming "Subject-**and**-Object", which is for distillation and will be further discussed in Section 3.3. We also curate synthetic data from Unity3D to test its effectiveness as shown in Section 4.3.

### 3.3 NETWORK ARCHITECTURE

Figure 3 shows the overall architecture of our method. This method adopts the foundational structure of distillation learning Hinton (2015), consisting of a student network and a teacher network.

#### 3.3.1 STUDENT NETWORK

As mentioned in Section 1, we optimize a rarely used token $v*$ to capture the visual interactive action, then insert this token as predicate into the inference prompt to guide image generation with interaction actions. The optimization objective is shown below:

$$v_* = \arg\min_v \mathbb{E}_{z \sim t, z_0, \epsilon, c} \left[ \| \epsilon - \epsilon_\theta \left( z_t, t, c(y) \right) \|_2^2 \right], \tag{4}$$

where $v*$ is the token to represent the interaction action, $y$ is the textual condition, a "Subject-$v*$-Object" structured prompt.

A straightforward method is to optimize it as Textual Inversion Gal et al. (2022) with a denoising loss as Eq. 3.1, but only using denoising loss is prone to overfitting to the synthetic data. For example, as shown in Figure 2, the generated image contains the style of synthetic data and the color of the agents. A common way to disentangle these object appearance and style is to follow personalization methods like DreamBooth Ruiz et al. (2023), but its prior preservation loss requires additional real images to prevent overfitting, which is hard to obtain. ReVersion focuses on modeling the preposition relation (in, of, on, etc) of objects and can disentangle object appearance and style without additional images with a loss like Info-NCE Oord et al. (2018). Similarly, we use a contrastive loss to steer the interactive action embedding towards the interactive action region in the text embedding space:

$$L_{\text{steer}} = -\log \frac{\sum_{l=1}^{L} e^{v^\top \cdot P_i^l / \gamma}}{\sum_{l=1}^{L} e^{v^\top \cdot P_i^l / \gamma} + \sum_{m=1}^{M} e^{v^\top \cdot N_i^m / \gamma}}, \tag{5}$$

where $\gamma$ is the temperature parameter, $P_i = \left\{ P_1^1 ..., P_i^L \right\}$ refers to positive samples, from the embeddings of diverse interactive actions (we collect over 150 words and phrases), and $N_i = \left\{ N_1^1 ..., N_i^M \right\}$ refers to negative samples, embeddings from other Part-Of-Speech of the input tokens.

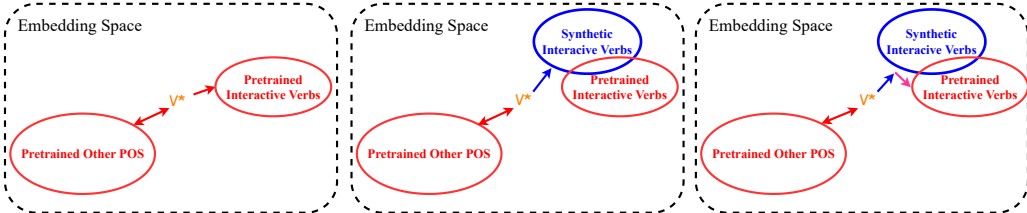

Figure 4: Diagrams of embedding space, POS refers to Part-Of-Speech, red circle represents embedding pretrained on real data, and blue circle is synthetic embedding. Left shows on real data, how $v*$ is steered to an interactive action space, however, as mentioned in Section 1 there are rare physical real images. As the middle shows, due to the domain gap, synthetic embedding deviates from the pretrained real embedding. During optimization, student network will overfit to the synthetic data, and $v*$ is pushed to an area that cannot invert real actions in real images. As shown in the rightmost, the distillation loss (pink) will not push $v*$ too much and then let $v*$ reach real data space.

However, this steering process will result in a misalignment problem as shown in Figure 2, where subject or object will be thrown randomly. As shown in Figure 4, we hypothesize that unlike methods fine-tuning on real data which has identical distribution as the pretrained data, fine-tuning on synthetic data faces the problem that synthetic visual embedding is deviated from real visual embedding, and does not align with textual embeddings. Simply contrasting with other Part-of-Speech will not steer $v*$ to the interactive action space for inference in real data generation. To alleviate this problem, we propose a teacher network with a distillation loss to recify the steering process.

### 3.3.2 TEACHER NETWORK

As mentioned in Section 1, when dealing with interactive scenarios, vanilla stable diffusion models can precisely depict both the subject and the object, although fail to represent their spatial relationship. At the same time, the student network faces the misalignment problem that the subject or object will be lost randomly. This inspires us to integrate the advantage of the vanilla stable diffusion model which does not lose objects into our student network to address the misalignment problem.

Specifically, for identical noisy input, we assign different textual conditions to the student network and teacher network. The student network $\epsilon_\theta$ is conditioned on "Subject-$v*$-Object" structured textual prompt $y$ as previously mentioned. The teacher network, which is a frozen copy of the student network, is conditioned on a "Subject-**and**-Object" structured textual prompt $y_{and}$. At each training step, $y$ and $y_{and}$ are randomly sampled from two separate prompt groups (typically, 10-12 textual prompts) which are paired with each image. As shown in Figure 3, we adopt the concept of knowledge distillation by transforming the entire network structure into a distillation paradigm.

We use a classical distillation loss $\mathcal{L}_{distill}$ to compute the difference between the student network output and teacher network output, and minimizing this loss will guide the student network to depict both the subject and object, preventing missing either component of the textual prompt:

$$\mathcal{L}_{distill} = \|\epsilon_{\theta_{teacher}}(z_t, t, c_{\theta_{teacher}}(y_{and})) - \epsilon_\theta(z_t, t, c_\theta(y_{and}))\|_2^2 \tag{6}$$

$$= \|(\epsilon_{\theta_{teacher}}(z_t, t, c_{\theta_{teacher}}(y_{and})) - \epsilon) - (\epsilon_\theta(z_t, t, c_\theta(y_{and})) - \epsilon)\|_2^2 \tag{7}$$

Since $\theta_{teacher}$ and $\theta$ are initialized from the same weights, their initial output will be similar, while different from $\epsilon$. If the student network $\epsilon_\theta$ overfit on synthetic data $\|\epsilon_\theta - \epsilon\|_2^2$ will be small, and $L_{distill}$ will be large. Therefore we can regard $L_{distill}$ as a penalty term to prevent overfitting to the synthetic data. We present a visualization diagram as shown in Figure 4 for further explaination.

Naturally, distillation loss should not overwhelm the denoising loss and steering loss. We leverage $\lambda_{steer}$ and $\lambda_{distill}$ to balance them. The total loss is formulated as:

$$\mathcal{L} = \mathcal{L}_{denoise} + \lambda_{steer}\mathcal{L}_{steer} + \lambda_{distill}\mathcal{L}_{distill}. \tag{8}$$

To show what $v*$ learns, we visualize the cross-attention map in the student network as Figure 5.

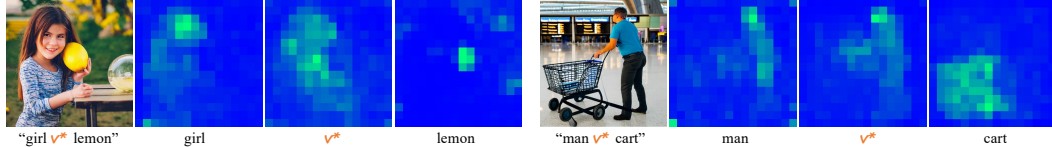

"girl **v*** lemon"   girl   **v***   lemon   "man **v*** cart"   man   **v***   cart

Figure 5: On the left, after fine-tuning on synthetic data with the "pick" action, $v*$ corresponds to the area connecting subject (girl) and object (lemon). On the right, after fine-tuning on synthetic data containing "push" action, $v*$ aligns with the area between subject (man) and object (cart).

## 4 EXPERIMENTS

### 4.1 EXPERIMENT SETTING

**Comparison Methods.** We select the following baselines to evaluate our methods effectiveness: 1) Stable Diffusion v-1.5, since there is no ground-truth textual description, we use natural language that can best describe sample images to replace the $v$ token. 2) DreamBooth requires generating 200 class-preservation images, and there will be a large domain gap between the generated real images and synthetic data, so we do not fine-tune it on the synthetic data but on selected real data among large number of generated images 3) Textual Inversion, ReVersion and Our method, all fine-tune on our curated synthetic data, based on vanilla Stable Diffusion v-1.5 model.

**Implementation Details.** By default, we set $\lambda_{distill} = 0.2$. We use the AdamW Kingma & Ba (2014) optimizer with a learning rate of 2.5e-4, while the training takes 3000 steps and training batch size is 2. We use 50 steps of the DDIM Song et al. (2020) sampler with a guidance scale of 7.5. All images are at a resolution of 512×512. All experiments are conducted on one NVIDIA A100 GPU.

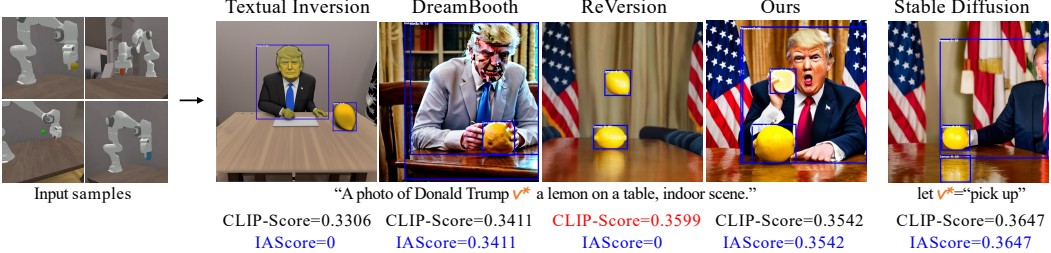

Input samples | Textual Inversion | DreamBooth | ReVersion | Ours | Stable Diffusion

"A photo of Donald Trump **v*** a lemon on a table, indoor scene."   let **v***="pick up"

CLIP-Score=0.3306   CLIP-Score=0.3411   CLIP-Score=0.3599   CLIP-Score=0.3542   CLIP-Score=0.3647
IAScore=0   IAScore=0.3411   IAScore=0   IAScore=0.3542   IAScore=0.3647

Figure 6: **(for ReViewer ro2Y's W2, ESr5' W1)** Example to illustrate CLIP-Score alone fails to evaluate the interactive action fairly. While our proposed IAScore could alleviate this unfairness.

**Evaluation Metric.** As mentioned in Section 1, existing conditional encoders does not align the textual and visual features of interactive action well, naturally, directly using these alignment evaluation metrics like CLIP-Score Radford et al. (2021) cannot obtain fair evaluation, as shown in Figure 6.

We assumed that most interactive actions require the subject to overlap with the object. Therefore, we propose a new metric $IAScore$ (Interactive Action Score): using an object detection method to calculate the Intersection over Union (IoU) score between the subject and object, and only samples with an IoU score greater than zero will be counted for textual-visual alignment evaluation. This approach helps eliminate images that score well but do not exhibit interactive action.

$$IAScore = \frac{1}{N} \sum_{i}^{N} sgn(IoU_i) \cdot CLIPScore_i \tag{9}$$

### 4.2 MAIN COMPARISONS

For evaluation, we have build 136 unique inference templates for each interactive action. The subjects used include a diverse range of people with distinct characteristics, from specific types of individuals to celebrities, as well as animals. Objects in these templates vary from non-articulated items (e.g., boxes) to articulated ones (e.g., refrigerators)in real life. This extensive collection enables a comprehensive assessment of performance across broad types of interactive actions.

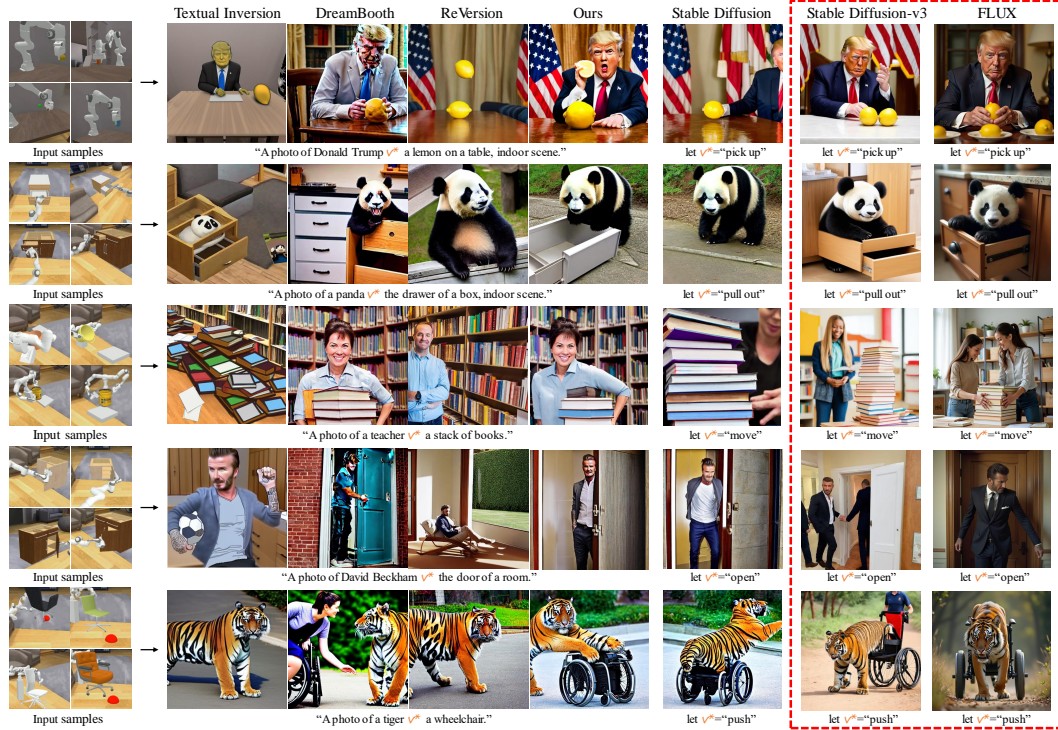

Figure 7: **(for Reviewer WdtC' W1, Esr5' Q4)** Main comparison results. Left are samples of the synthetic data used for fine-tuning by the four methods shown in the middle. Images on the right are results of Stable Diffusion v1-5 (used as main backbone), Stable Diffusion v3 Sauer et al. (2024) and FLUX as references. They validate our method's best performance for interactive action generation.

| Methods | $IAScore$ |
|---|---|
| Stable Diffusion | 0.2296 |
| DreamBooth | 0.2637 |
| Textual Inversion | 0.2021 |
| ReVersion | 0.1858 |
| Ours | **0.3386** |

Table 1: Quantitive Results

| Methods | Entity | Action | Overall |
|---|---|---|---|
| Stable Diffusion | 10.96% | 4.11% | 5.48% |
| DreamBooth | 9.59% | 5.48% | 2.74% |
| Textual Inversion | 2.74% | 4.5% | 0.00% |
| ReVersion | 6.85% | 1.37% | 2.74% |
| FLUX | **45.21%** | 21.92% | 30.14% |
| Ours | 26.03% | **64.38%** | **58.90%** |

Table 2: **(for ReViewer WdtC)** Qualitative Results by Human Evaluators adding FLUX.

**Qualititive Comparison**. Figure 7 illustrates the qualitative comparison of all methods involved. It can be observed that although natural text descriptions are provided, the interactive actions generated by Stable Diffusion are wrong (for example, the panda and teacher). Textual Inversion overfits on the synthetic data's appearance and style (for example, the desk of Trump is similar to the given example images). DreamBooth only partially captures the interactive action. ReVersion can disentangle the synthetic data's domain features, but it faces a misalignment problem: subjects or objects disappear randomly. Our method can learn from the example images and generate images authentic to the input prompt with reasonable spatial relation and object appearance.

**Quantitive Comparison**. We evaluate comparison methods with our proposed metric, as shown in Table 1. We conduct a survey involving 73 human evaluators to assess 50 groups of comparison images. Each group contains images generated by comparison methods. Along with the generated images, evaluators receive 1) the corresponding synthetic image showcasing a specific interactive action and 2) a natural language description of the synthetic image.

Human evaluators vote for the best-generated image based on three metrics: 1) Subject and Object (Entity) Accuracy. Evaluators judge whether the subjects and objects are generated accurately without any omissions, distortions, or deformations. 2) Interactive Action Accuracy. Evaluators assess whether the subjects and objects in the generated images accurately represent the interactive

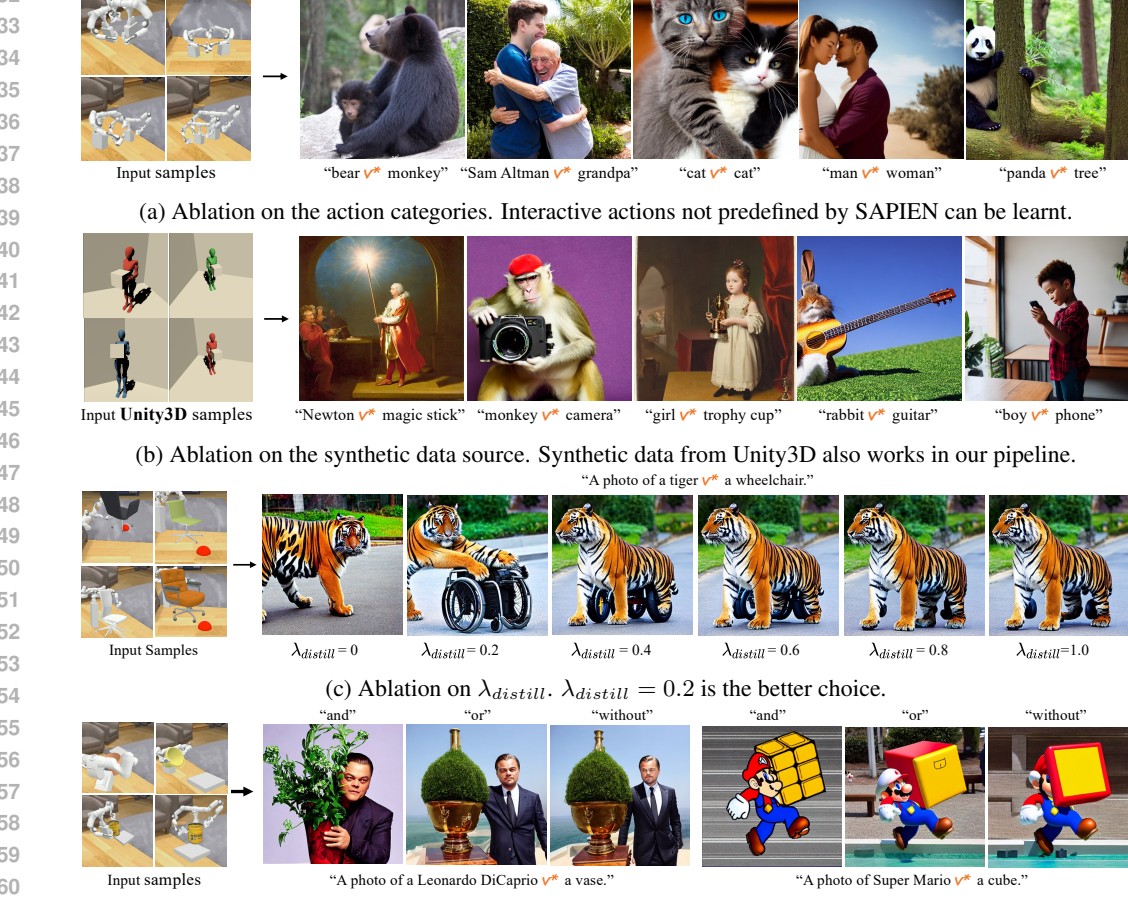

(a) Ablation on the action categories. Interactive actions not predefined by SAPIEN can be learnt.

(b) Ablation on the synthetic data source. Synthetic data from Unity3D also works in our pipeline.

(c) Ablation on $\lambda_{distill}$. $\lambda_{distill} = 0.2$ is the better choice.

(d) Ablation on the preposition. "and" is the better preposition for the teacher network's conditional prompt.

Figure 8: Ablation Study from four aspects.

actions depicted in the exemplar images and whether the spatial information is correctly expressed. 3) Overall Quality. Evaluators rate the overall quality of the generated images. Even if the objects and their interactions are accurately generated, this metric checks if their fidelity is maintained.

As indicated in Table 2, our method has superior performance across all three quality metrics.

## 4.3 ABLATION STUDY

**Ablation on actions.** To assess the transferability of our method, we conduct an ablation study on the input sample group. The results indicate that our pipeline can effectively learn interactive actions not pre-defined by the SAPIEN platform as shown in Figure 8.

**Ablation on synthetic data source.** To further evaluate our method's transferability, we ablate the synthetic data source. From the graphics engine platform Unity3D, we manually synthesize interactive action images that seem physically reasonable, then apply them in our pipeline. The results in Figure 8 show that our method can be extended to different synthetic data.

**Ablation on $\lambda_{distill}$.** We ablate $\lambda_{distill}$ to further investigate its impact. As shown in Figure 8, when $\lambda_{distill} = 0$, the distillation branch takes no effect, and subjects or objects disappear randomly, which proves our distillation is necessary. What's more, $\lambda_{distill}$ should not be too large, if $\lambda_{distill} = 1.0$, the interactive action disappears, we hypothesize that is due to the distillation dominating the process. We recommend using empirically default value $\lambda_{distill} = 0.2$, which can cover most cases.

**Ablation on preposition word.** As previously mentioned, we typically use "and" as a preposition to link subjects and objects in the distillation branch's condition text. We replaced "and" with "or" and "without" to evaluate their effectiveness, as Figure 8 showing that "and" performs better.

## 4.4 ADDITIONAL EXPERIMENTS

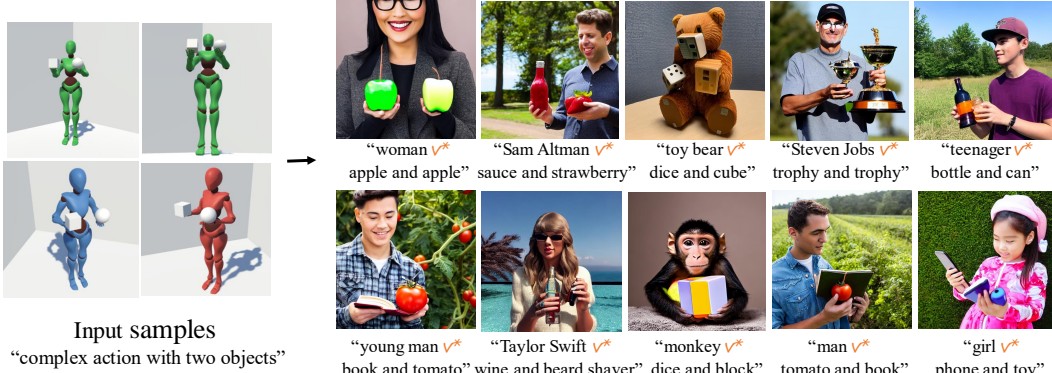

Figure 9: Additional experiments on a more complex Interactive Action with Multiple Objects.

**Experiments on more complex samples. (for Reviewer ro2Y' W1, gBhf' W1&Q1, W9Rr' W1&Q3)** As Figure 9 show, the generated results validate our pipeline's generalization ability.

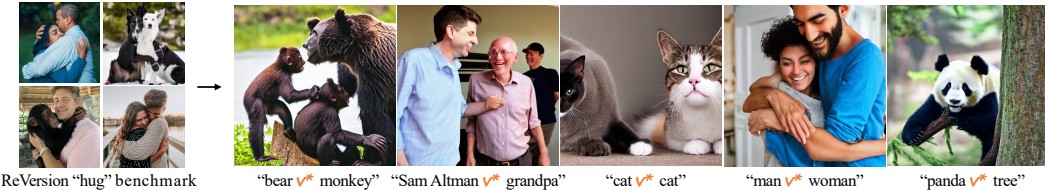

Figure 10: Additional experiments of ReVersion on 'hug' benchmark.

**Experiments of ReVersion. (for Reviewer W9Rr' Q1)** ReVersion primarily focuses on preposition words rather than interactive action words. To enable a fair comparison, we chose the only common action, "hug". In Figure 10, ReVersion demonstrates inaccurate physical positioning, performing worse than ours in Figure 8, likely due to the absence of multi-view information in the real data.

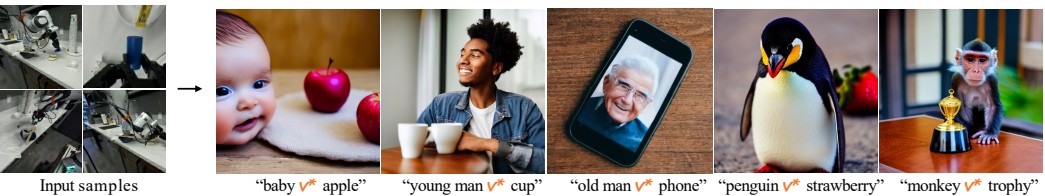

Figure 11: Additional experiments on robot manipulation dataset DROID Khazatsky et al. (2024).

**Experiments on DROID Dataset. (for Reviewer WdtC' W5)** As shown in Figure 11, we sampled "pick-up" action images from the DROID Khazatsky et al. (2024) dataset to adapt them to our pipeline. Experimental results reveal that fine-tuning on DROID struggles to capture interactive actions. This may be because its actions are challenging to recognize effectively.

## 5 CONCLUSION

In this work, we pioneer in generating images with interactive actions based on the text-to-image diffusion model. To address this challenge, we have curated a synthetic dataset enriched with controllable interactive actions rendered from a graphics engine. Based on the dataset, a novel fine-tuning scheme is proposed to improve the spatial and interactive awareness of the diffusion model. With the steering loss and distillation structure, our proposed method effectively addresses the problems of disentanglement and textual-visual misalignment and achieves state-of-the-art performance, marking synthetic data's potential in interactive-action image generation.

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

# A  APPENDIX

## A.1  CODE

Our code and dataset are available at this anonymous external link: `https://anonymous.4open.science/r/Interactive_Action-E033`

## A.2  MORE SYNTHETIC DATA

In the main text, we only shows several sampled input images, here we show more images by SAPIEN and images from Unity3D which are used for ablation study.

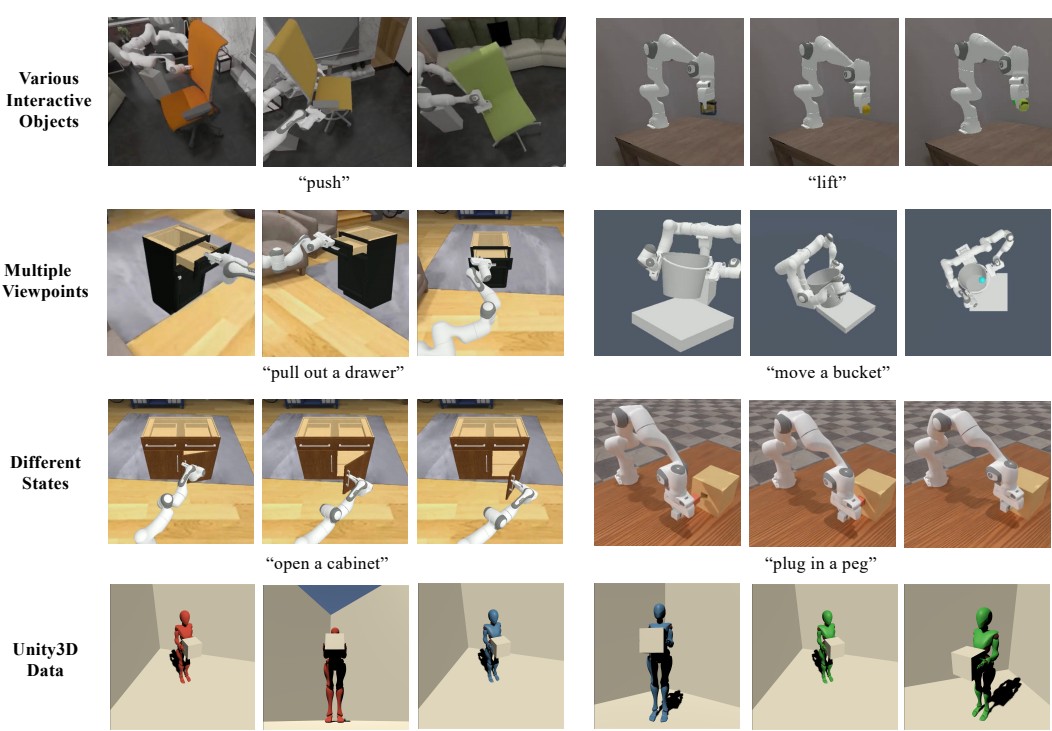

Figure 12: Examples of synthetic dataset built with SAPIEN engine. This dataset contains various interactive actions, various objects, multiple rendering viewpoints and different stateswhich offers sufficient physical knowledge. Note that here the name of the action is only for representation convenience since one action can correspond to several synonymous words the action "pick up the cube" can be replaced by "lift the cube", "open the door" can be replaced by "pull out the door".

## A.3  DATA GENERATION PIPELINE

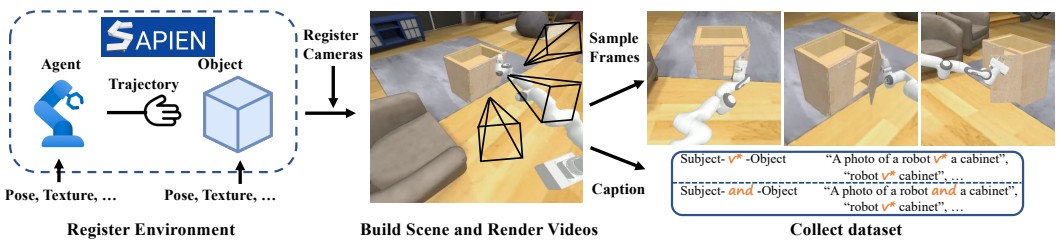

Figure 13: Synthetic data generation pipeline based on SAPIEN.

As can be seen from Figure 13, SAPIEN is a 3D realistic and physics-rich simulation platform. The platform provides an extensive library of pre-designed articulated robot agents, objects, and

habitat assets, enabling users to simulate realistic interactive environments. These robot agents are programmed to manipulate objects along trajectories sampled from a predefined set. Under the guidance of these sampled trajectories, the agents can efficiently navigate and interact with objects.

Users have the flexibility to select specific objects for the agents to interact with, enhancing the customization of the simulation. The habitat scenes are meticulously crafted to offer a realistic background, adding depth and authenticity to the simulations. This realism is further bolstered by user-defined cameras, which allow for the capture of the interactive actions from multiple viewpoints. To optimize the observational quality of these interactions, users can finely adjust the coordinates of both the agents and objects, as well as the camera angles. This adjustability ensures that users can obtain clear and precise observations of the interactive actions.

After the image data obtained, we caption them with two kinds of textual prompts: "Subject-$v*$-Object" structure and "Subject-and-Object" structure. Thus we obtain paired training data.

### A.4 LIMITATIONS AND BROADER IMPACT

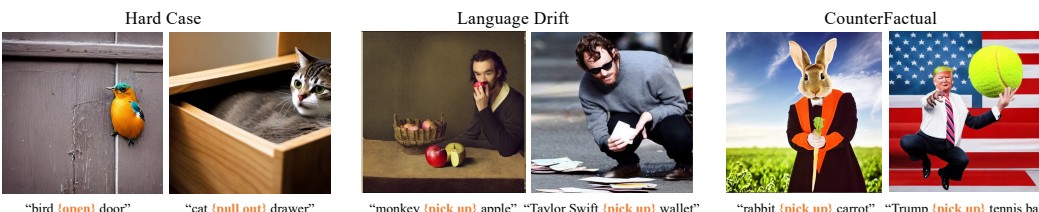

Figure 14: Examples of our failure cases. The leftmost represents that some animals are hard to finish some interactive tasks, the middle shows that sometimes subject may not match with the textual prompts, the rightmost indicates that although generated images can satisfy the interaction requirement, there are still some counterfactual results.

**Limitation.** As shown in Figure 14, our work can only activate the inherent knowledge within stable diffusion models but struggles to incorporate new knowledge. This limitation is influenced by the pretrained stable diffusion models, which vary in effectiveness depending on the subject. For instance, it is challenging for the model to depict "an otter picking up objects", likely because it has not been exposed to many images of otters performing such actions. In contrast, depicting a monkey performing similar tasks is much easier due to the model's frequent exposure to such images. This reflects a bias in stable diffusion.

Moreover, the output is occasionally not well-aligned with the text, introducing irrelevant words and objects. Additionally, the precision, especially concerning angles, is insufficient.

**Potential Negative Societal Impacts.** The entity relational composition capabilities of our method could be applied maliciously on real human figures.

### A.5 ADDTIONAL RESULTS

**(for Reviewer WdtC)** We conduct experiments on FLUX with multiple random seeds, and as shown in Figure 15, generating interactive-action images is still challenging.

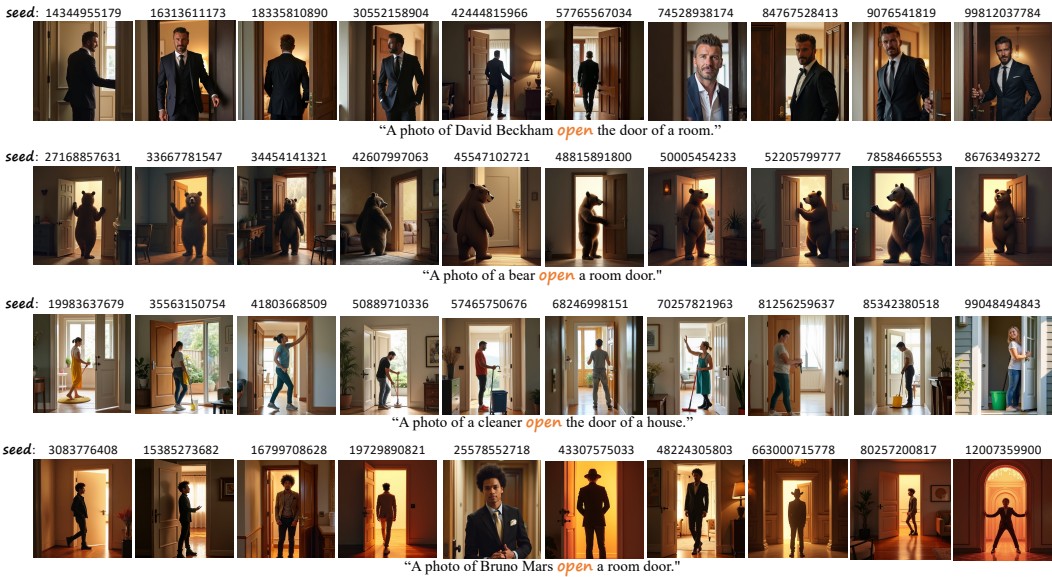

Figure 15: **(for Reviewer WdtC)** FLUX results with corresponding multiple random seeds and prompts.

