# OpenReview forum: "Interactive-Action Image Generation via Synthetic Physical Priors"
_ICLR.cc/2025/Conference — Submitted to ICLR 2025_

### Official Review · Reviewer_W9Rr · 2024-11-01

**Soundness:** 1
**Presentation:** 1
**Contribution:** 2
**Rating:** 3
**Confidence:** 5

**Summary:**

This paper addresses the issue of text-to-image generation models struggling to accurately depict interactive actions (e.g., "a person picks up a cup"). To address this, the authors generate a synthetic dataset enriched with physical constraints using a graphics engine. They train the model through a teacher-student network with distillation learning, enhancing the model's ability to capture spatial and interactive relationships. The approach reduces alignment issues between subjects and objects, improving realism in generated images.

**Strengths:**

* The distillation-based method is a straightforward yet reasonable approach.
* The authors address a challenging task effectively.
* They propose a new metric for evaluation.

**Weaknesses:**

* Limited Interaction Complexity: The paper focuses on single-object manipulation, which seems somewhat limited in scope.
* The qualitative results seem somewhat lacking. It seems this issue should be addressed through the proposed distillation method.

**Questions:**

* In the paper, they compare performance with ReVersion fine-tuned on their dataset, which is fair; however, the original ReVersion paper also demonstrates strong performance. For a more comprehensive comparison, the authors should include additional experimental results using the original (non-fine-tuned) ReVersion model.

* The ablation study shows that 'and' improves semantic understanding, while 'or' enhances visual representation. What is the reason for this conflict?

* Can the model perform in more complex interactions, such as multi-object manipulation or tool use (e.g., hammering a nail, tightening a screw with a screwdriver)?

---

> ### Author Response · Authors · 2024-11-23
> **Official Response to Reviewer W9Rr**
>
> We thank the reviewer for the valuable comments and seek to address reviewer's concerns below:
>
> ****
>
> >**W1** Limited Interaction Complexity: The paper focuses on single-object manipulation, which seems somewhat limited in scope.
>
> **R1**: Please see **Figure 9** of our updated version, in which we have added an experiment for complex actions, and our method reveals good performance.
>
> ****
>
> >**W2** The qualitative results seem somewhat lacking. It seems this issue should be addressed through the proposed distillation method.
>
> **R2**: Sorry we did not clearly understand your issue about "lacking", can you elaborate on your point? We will update more qualitive results through our anonymous repostory link in the Appendix we uploaded during first submission, if you mean that you expect to view more results.
>
> ****
>
> >**Q1** In the paper, they compare performance with ReVersion fine-tuned on their dataset, which is fair; however, the original ReVersion paper also demonstrates strong performance. For a more comprehensive comparison, the authors should include additional experimental results using the original (non-fine-tuned) ReVersion model.
>
> **R3**: Thanks for your suggestion, we have added ReVersion's results, and please see **Figure 10** of our updated version. The results show that ReVersion perform inferior to our method in interactive action aspect.
>
> ****
> >**Q2** The ablation study shows that 'and' improves semantic understanding, while 'or' enhances visual representation. What is the reason for this conflict?
>
> **R4**: This ablation study reveals an uncommon result where the visual performance of 'and' is slightly lower than 'or.' Overall, their visual representations show minimal differences, but the 'and' semantics demonstrate greater accuracy.
>
> ****
>
> >**Q3** Can the model perform in more complex interactions, such as multi-object manipulation or tool use (e.g., hammering a nail, tightening a screw with a screwdriver)?
>
> **R5**: Please refer to Figure 9 in our updated version, where we demonstrate a more complex action with multi-object manipulation, and our method produces visually satisfactory results.
>
> For scenes like hammering a nail, we are unable to show results due to the limitations of our professional skills in rendering them in a 3D graphics engine. However, we believe that if treating such scenes as a video and applying our techniques to video generation models, we could synthesize these complex scenarios effectively.

---

> ### Author Response · Authors · 2024-12-02
> **A Gentle Reminder of Feedback**
>
> Dear Reviewer W9Rr,
>
> We deeply value your initial feedback and understand that you may have a busy schedule. However, we kindly request that you take a moment to review our responses to your concerns. Any feedback you can provide would be greatly appreciated. We are also available to address any additional questions before the rebuttal period concludes.
>
> Sincerely,
>
> The Authors

---

> > ### Comment · Reviewer_W9Rr · 2024-12-02
> >
> > I reviewed the feedback from other reviewers and the authors' responses and appreciate their efforts to enhance the work. However, I believe further development, such as a quantitative ablation study or additional qualitative comparisons, is still needed. Thus, I will maintain my original score.

---

> ### Author Response · Authors · 2024-12-03
>
> Thank you for your time and thoughtful feedback.
> >**Q1** More quantitative ablation results.
>
> **R1:** Due to the constraints of computational resources, we were unable to include all the additional experiments during the rebuttal period. Instead, we prioritized utilizing these resources to focus on other experiments as you requested, such as more complex and multi-object action as **Figure 9** and incorporating additional comparison baselines such as and ReVersion in **Figure 11**, which further validate the effectiveness of our method.
>
> In the meantime, we believe the visual results have been enough to effectively demonstrate the robustness of our approach (as shown in Ablation a and b) and the most favorable experimental settings (as illustrated in Ablation c and d).
>
> We will certainly consider your request if circumstances allow in the future. We sincerely hope that our method and its contributions to this challenging problem provide valuable insights for you. Thank you for your thoughtful review and consideration.
> ****
>
> >**Q2** More qualitative comparison results.
>
> **R2:** In your initial response during the rebuttal period, you mentioned that **qualitative results** are lacking, which we found unclear. To gain a better understanding, we requested clarification but did not receive a response. Considering that you may have a busy schedule, we took the initiative to put more our qualitative results via the provided anonymous link.
>
> However, it now appears that your concern may have been about the **qualitative comparisons** being insufficient. To address this at now, we kindly suggest referring to the User Study results in **Table 2**, which include multiple groups of comparisons. These results provide further evidence that our method outperforms others. We appreciate your feedback and hope this addresses your concerns.

---

### Official Review · Reviewer_ESr5 · 2024-11-03

**Soundness:** 3
**Presentation:** 3
**Contribution:** 3
**Rating:** 6
**Confidence:** 4

**Summary:**

The paper presents a method to enhance diffusion-based text-to-image generation for interactive actions by synthesizing a dataset with physical priors using a graphics engine. It introduces a distillation-based fine-tuning approach that improves understanding of interactive actions and reduces misalignment. Experiments show that this method achieves state-of-the-art results and expands the use of synthetic data in generating interactive images.

**Strengths:**

1. This paper propose a novel method that synthesizes a dataset enriched with physical priors using a graphics engine.
2. This paper introduce a distillation-structured fine-tuning method using SD.
3. Qualitative results are good compared to the baselines.

**Weaknesses:**

1. It is challenging to evaluate this as a new approach. However, the authors propose a new metric, the IAS score, and evidence is needed to demonstrate that this metric in the evaluation.
2. There is a lack of quantitative analysis for the ablation studies. (such as, preposition word ..)
3. The authors released the code in the supplementary materials. Is it possible to view more samples of the dataset or generated results?
4. I am curious about if SD-XL or other models were used instead of SD1.5.

**Questions:**

Please refer to the weaknesses.

---

> ### Author Response · Authors · 2024-11-23
> **Official Response to Reviewer ESr5**
>
> We thank the reviewer for the valuable comments and seek to address reviewer's concerns below:
>
> ****
>
> >**W1** It is challenging to evaluate this as a new approach. However, the authors propose a new metric, the IAS score, and evidence is needed to demonstrate that this metric in the evaluation.
>
> **R1**: This IAScore is an intuitive design that is logically sound, aligns with the definition of high-quality interactive actions, and is consistent with the user study results. And we have updated **Figure 6** to further explain its ability. And we will look into the limitations of IAScore and try to improve it in the future.
>
>
> ****
>
> >**W2** There is a lack of quantitative analysis for the ablation studies. (such as, preposition word ..)
>
> **R2**: Due to the large number of ablation experiments, we did not evaluate quantitative metrics. However, the visual results are sufficient to determine which outcomes are preferable.
>
> ****
>
> >**W3** The authors released the code in the supplementary materials. Is it possible to view more samples of the dataset or generated results?
>
> **R3**: Yes, thanks for your interest in our work, we will update more samples and results in our released anonymous repository. And you can also view **Figure 9** of our updated version.
>
> ****
>
> >**W4** I am curious about if SD-XL or other models were used instead of SD1.5.
>
> **R4**: Yes, simply adjusting the resolution allows methods like SD-XL to be applied in our pipeline. While more powerful T2I baseline models like SD3 or FLUX performs poorly in interactive action image generation as shown in **Figure 1 and Figure 7** of our updated version.

---

> > ### Comment · Reviewer_ESr5 · 2024-11-24
> >
> > I saw the reviews from other reviewers and the authors' responses. Only some of my concerns have been addressed, so I will maintain my original score. I appreciate the authors' best efforts in their responses, but the remaining concerns need to be further addressed.

---

### Official Review · Reviewer_WdtC · 2024-11-03

**Soundness:** 2
**Presentation:** 2
**Contribution:** 2
**Rating:** 3
**Confidence:** 4

**Summary:**

This paper proposes a method to improve the capacity of text-to-image diffusion models to generate interactions by fine-tuning the model on synthetic data. To address the domain gap between the synthetic data and real images, the author developed a distillation approach to preserve the performance of the mode in generating the realistic style while still learning the interactions from the synthetic data.

**Strengths:**

This paper studies the interesting and important problem that visual generative models lack the proper knowledge about the physical world and are not able to properly generate physical-plausible actions. The authors propose reasonable fine-tuning-based methods and collect synthetic data from a simulator to address this problem. The authors show that the existing methods do not work well due to the synthetic-real domain gap. Ablation studies are performed to help understand the function of each component.

**Weaknesses:**

My major concern about this paper is its motivation to improve the ability of text-to-image models to generate human/animal-object interactions. With the improved performance of the base generation, I have been seeing this as less and less of a problem. In many of the examples shown in the papers where SD1.5 fails to generate correct interactions, the latest models, like SD3 and Flux, can do pretty well. That being said, I do believe that there are many actions that cannot be accurately described by the language. However, I don't see current methods as a solution to that, as there are cases where the learned motions are not clearly matched with the training images. For example, in the third row of the teaser image, the robot arm picks up the objects with two hands, while the generated images mostly contain a subject holding an object with one hand. Also, I also have the following concerns:

1. There are some conflicts between the different losses, i.e., the distillation loss would conflict with learning the interaction through the synthetic data, which is also verified by the ablation study in Figure 8 (c). The optimal trade-off between two losses may not be the same for different interactions.

2. The issue with overfitting the synthetic data has been studied, and the function of the proposed distillation loss is quite similar to the class-specific prior preservation loss. Although the authors claim that real data is not common, robot manipulation datasets like BridgeDatav2 and DROID datasets do exist.

3. There exists another gap that the interaction between the robot arm and objects is quite different from how humans interact with objects - the robot arm typically does not have a similar structure compared with human hands. Instead, one could just collect more human-object interaction data. And there do exist such datasets, like Epic Kitchen, Hands23, and Ego4D.

**Questions:**

I do not have additional questions.

---

> ### Author Response · Authors · 2024-11-23
> **Official Response to Reviewer WdtC (Part 1/2)**
>
> We appreciate your efforts to inverstigate the motivation of our paper from multiple perspectives.
>
> ****
>
> >**W1** My major concern about this paper is its motivation to improve the ability of text-to-image models to generate human/animal-object interactions. With the improved performance of the base generation, I have been seeing this as less and less of a problem. In many of the examples shown in the papers where SD1.5 fails to generate correct interactions, the latest models, like SD3 and Flux, can do pretty well.
>
> **R1**: Please see **Figure 1** and **Figure 7** of our updated version. As the results show, SD3 and FLUX enhance the appearance fidelity of individual objects, but the interactive actions between them remain far from satisfactory.
>
> ****
>
> >**W2** That being said, I do believe that there are many actions that cannot be accurately described by the language. However, I don't see current methods as a solution to that, as there are cases where the learned motions are not clearly matched with the training images. For example, in the third row of the teaser image, the robot arm picks up the objects with two hands, while the generated images mostly contain a subject holding an object with one hand.
>
> **R2**: While our results are not always prefect, they demonstrate significant improvement over existing baselines. Our key contribution is demonstrating the potential of synthetic graphics data in tackling this challenging problem.
>
> ****
>
>
> >**W3** There are some conflicts between the different losses, i.e., the distillation loss would conflict with learning the interaction through the synthetic data, which is also verified by the ablation study in Figure 8 (c). The optimal trade-off between two losses may not be the same for different interactions.
>
> **R3**: As mentioned in our paper line 329, line 472 (**updated version line 323, line 482**) and in our released code, we set $0.2$ as the default value. In practice, $\lambda_{distill} = 0.2$ is empirically the best choice, which can cover most scenarios, and we do **NOT** need to optimize a trade-off for each example.
>
> ****
>
> >**W4** The issue with overfitting the synthetic data has been studied, and the function of the proposed distillation loss is quite similar to the class-specific prior preservation loss.
>
> **R4**: They are **NOT** similar. The two loss functions differ significantly in the following aspects:
>
> 1) **Usage condition**. Unlike the class-specific prior loss used in DreamBooth, which requires generating approximately 100-200 images offline or sourcing data from an external dataset, our loss function eliminates the need for such pre-processing steps.
>
> 2) **The domain gap problem**. The class-specific prior loss in DreamBooth struggles with significant domain gaps.
>
> Since it requires Stable Diffusion to generate images similar to the dataset for fine-tuning, Stable Diffusion cannot generaet such synthetic data, then this loss's performance will fall down.
>
> 3) **Practical performance**. As our experiments showed in **Figure 7** and **Table 1,2**, DreamBooth with the class-specific prior loss does not produce good results, while our method consistently yields excellent outcomes.

---

> ### Author Response · Authors · 2024-11-23
> **Official Response to Reviewer WdtC (Part 2/2)**
>
> > **W5** Although the authors claim that real data is not common, robot manipulation datasets like BridgeDatav2 and DROID datasets do exist.
>
> **R5**: Please see **Figure 11** of our updated version. These real dataset demonstrate poor results.
> We believe that these real-world datasets are not well-suited to our method for the following reasons:
>
> 1. **Limited Viewpoint Flexibility:** The datasets lack the flexibility to render multi-view images freely, making it challenging to obtain sufficient physics priors.
>
> 2. **Cluttered Imagery:** Many images in these datasets are too cluttered, making it difficult to isolate and focus on specific interactive actions and objects.
>
> 3. **Labor-Intensive Data Collection:** Gathering relevant data is time-consuming and requires meticulous manual selection, making it far less convenient compared to using synthetic data."
>
> ****
>
> >**W6** There exists another gap that the interaction between the robot arm and objects is quite different from how humans interact with objects - the robot arm typically does not have a similar structure compared with human hands. Instead, one could just collect more human-object interaction data. And there do exist such datasets, like Epic Kitchen, Hands23, and Ego4D.
>
> **R6**: While hand-focused tasks are an important research area, our work emphasizes the **_entire body_**. Datasets such as Epic Kitchen, Hands23, and Ego4D primarily focus on hand-object interactions and often lack comprehensive information about the full human body.
>
> Our objective is to generate complete human-object interactions, capturing the full body’s pose and dynamics rather than isolating hand-object relationships. As such, these datasets are not well-suited to our requirements.

---

> > ### Comment · Reviewer_WdtC · 2024-11-25
> >
> > I appreciate the authors' efforts in the additional experiments and responses. Though some of my concerns are addressed by the responses, I'm still unconvinced by the proposed solution to this problem:
> >
> > 1. Given the examples in Figures 1 and 7, I think they demonstrate that better text-to-image models face much less of an issue in generating human-object interactions. For example, the Flux "pick-up" example in Figure 7 looks good to me, which failed in the originally compared baselines. In addition, I tried the other prompts where Flux is reported to fail. However, I still observed a decent success rate in some prompts, e.g., "opening doors." I think a more convincing comparison is across multiple random seeds with human evaluation to show a clear improvement over these methods.
> >
> > 2.  **R6**: There are also human-object interactions with the full-body pose captured. For example, FullBodyManipulation [1], EgoExo4D [2], and EgoBody [3].
> >
> > [1] Li, Jiaman, Jiajun Wu, and C. Karen Liu. "Object motion guided human motion synthesis." ACM Transactions on Graphics (TOG) 42.6 (2023): 1-11.
> >
> > [2] Grauman, Kristen, et al. "Ego-exo4d: Understanding skilled human activity from first-and third-person perspectives." Proceedings of the IEEE/CVF Conference on Computer Vision and Pattern Recognition. 2024.
> >
> > [3] Zhang, Siwei, et al. "Egobody: Human body shape and motion of interacting people from head-mounted devices." European conference on computer vision. Cham: Springer Nature Switzerland, 2022.

---

> > > ### Author Response · Authors · 2024-11-26
> > > **Official Response to Reviewer WdtC (4/4)**
> > >
> > > We have updated the User Study, adding a comparison with FLUX in Table 2. It indicates that FLUX's entity quality are favored by users due to its improvement, but when considering interactive actions, users still prefer ours.

---

> ### Author Response · Authors · 2024-11-25
> **Official Response to Reviewer WdtC (3/3)**
>
> >Given the examples in Figures 1 and 7, I think they demonstrate that better text-to-image models face much less of an issue in generating human-object interactions. For example, the Flux "pick-up" example in Figure 7 looks good to me, which failed in the originally compared baselines. In addition, I tried the other prompts where Flux is reported to fail. However, I still observed a decent success rate in some prompts, e.g., "opening doors." I think a more convincing comparison is across multiple random seeds with human evaluation to show a clear improvement over these methods.
>
> **R1:** The updated Figure 1 highlights that recent text-to-image models **still struggle** in generating accurate interactive images. For instance, in the "girl pulling out a drawer" example, the spatial relationship between the girl and the drawer remains incorrect, similar to the baseline. Similarly, in the "magician moving a stone" example, the magician's hand fails to make contact with the stone. These examples underscore the ongoing difficulties in achieving realistic and coherent interactive image generation.
>
> In Figure 7's "Trump picking up a lemon" example, the model only depicts him touching the lemon on the table, failing to illustrate the actual "picking up" action. Furthermore, the sucess rate is still low. To assess FLUX's performance in generating "opening doors" scenarios, we conducted additional experiments detailed in **Figure 15 of updated Appendix**. As demonstrated by the examples of the cleaner and Bruno Mars, the issue remains unresolved by FLUX, with a consistently **low success rate**. And User study results adding FLUX evaluation results will soon be updated.
>
> ****
>
>
> >**R6**: There are also human-object interactions with the full-body pose captured. For example, FullBodyManipulation [1], EgoExo4D [2], and EgoBody [3].
>
> **R2:** We are uncertain whether FullBodyManipulation [1] supports rendering **articulated** objects, such as drawers and cabinets in SAPIEN, as its released data appears to include only non-articulated and portable objects like "clothes stand," "floor lamp,""large box" and "plastic box." It seems that FullBodyManipulation is a generative action generation method capable of generating SMPL humans and objects scenarios, but its robustness may not be guaranteed. Its results are similar to what we built with Unity3D, which could potentially be compatible with our pipeline. We appreciate your effort in searching for this dataset, if we find it convenient for rendering multi-view interactions, we will consider adapting it into our pipeline and citing it as a reference.
>
> Similar to DROID, real datasets like EgoExo4D [2] and EgoBody [3] lack the flexibility to freely render multi-view images, making it hard to obtain sufficient physics priors. They also cannot cover all target interactive actions as users want. Additionally, gathering relevant data from these datasets is time-consuming and requires careful manual selection.
>
> [1] Li, Jiaman, Jiajun Wu, and C. Karen Liu. "Object motion guided human motion synthesis." ACM Transactions on Graphics (TOG) 42.6 (2023): 1-11.
>
> [2] Grauman, Kristen, et al. "Ego-exo4d: Understanding skilled human activity from first-and third-person perspectives." Proceedings of the IEEE/CVF Conference on Computer Vision and Pattern Recognition. 2024.
>
> [3] Zhang, Siwei, et al. "Egobody: Human body shape and motion of interacting people from head-mounted devices." European conference on computer vision. Cham: Springer Nature Switzerland, 2022.

---

### Official Review · Reviewer_gBhf · 2024-11-04

**Soundness:** 3
**Presentation:** 3
**Contribution:** 3
**Rating:** 8
**Confidence:** 3

**Summary:**

This paper first points out that existing diffusion models tend to struggle with verbs that describe relationships between objects. To address this, the authors propose a method for fine-tuning these diffusion models to better capture action-related information. They generate synthetic data representing actions using robot simulation model, SAPIEN, and train a special token following a textual inversion framework. To account for the domain gap between synthetic and real images, they employ a distillation learning approach for fine-tuning. The proposed method shows improved results compared to previous studies on the evaluation benchmark designed by the authors.

**Strengths:**

- The use of robot simulation to generate datasets for certain actions is an interesting and innovative approach.
- The paper clearly explains the challenges encountered during the development and the method is well described without ambiguity.
- The authors introduce new metric for evaluating the proposed task, and clearly present comparison results with previous studies.

**Weaknesses:**

- The proposed method is limited to simple actions that can be performed in robot simulations, such as picking up or pulling something. More complex actions like tickling or shaving cannot be represented in this way, making it difficult to train on broader action concepts.
- The use of distillation learning increases the training load compared to previous studies.

**Questions:**

Regarding the weakness mentioned above, how can a dataset be generated for more complex actions?

---

> ### Author Response · Authors · 2024-11-23
> **Official Response to Reviewer gBhf**
>
> We thank the reviewer for the valuable comments and seek to address reviewer's concerns below:
>
> ****
>
> >**W1** The proposed method is limited to simple actions that can be performed in robot simulations, such as picking up or pulling something. More complex actions like tickling or shaving cannot be represented in this way, making it difficult to train on broader action concepts.
>
>
> **R1**: Please see **Figure 9** of our updated version, where we synthesize a more complex action for our pipeline to learn and obtain visually satisfactory results. As long as the actions can be rendered by the graphics engine, our method could learn to generate them.
>
> ****
>
> >**W2** The use of distillation learning increases the training load compared to previous studies.
>
> **R2**: It increases, but marginally. In our actual tests, our baseline requires 18GB, while our method requires 21GB, resulting in only a 3GB increase.
>
> ****
>
> >**Q1** Regarding the weakness mentioned above, how can a dataset be generated for more complex actions?
>
> **R3**: This depends on the 3D rendering engine used. If it can render images of more complex actions, our method can leverage them for learning.

---

### Official Review · Reviewer_ro2Y · 2024-11-05

**Soundness:** 3
**Presentation:** 3
**Contribution:** 3
**Rating:** 6
**Confidence:** 4

**Summary:**

This work focuses on the interactive actions image generation based on pretrained text-to-image diffusion models, where a customized synthetic dataset with interactive actions is provided to finetune the pretrained model to improve the spatial and interactive action aware diffusion models.The finetune stage is akin to distillation procedure to address the problem of disentanglement and text-visiual misalignment. The experimental results demonstrate the effectiveness, especially the interactive actions between objects, of the proposed model.

**Strengths:**

- The paper is easy to follow. This work clearly identifies the weakness of generating interactive actions in current t2i models and proposes a effective solution to address the limitation.
- The method design is well-motivated, including the usage of curated dataset and the specific distillation training framework to take advantage of both pretrained prior and the new learned knowledge.

**Weaknesses:**

- While this work demonstrates the effectiveness on several interactive actions, it is unclear how well the method generalizes to a wider range of more complex interactions. Moreover, is the model capable of dealing with actions related to more than two objects.
- The motivation of IAScore is clear, however, its reliance on detection introduces potential errors. A more detailed analysis of the reliability and limitations of this metric would make it more credible.

**Questions:**

Please refer to the weakness part for more details.

---

> ### Author Response · Authors · 2024-11-23
> **Official Response to Reviewer ro2Y**
>
> We thank the reviewer for the valuable comments and seek to address reviewer's concerns below:
>
> ****
>
> > **W1** While this work demonstrates the effectiveness on several interactive actions, it is unclear how well the method generalizes to a wider range of more complex interactions. Moreover, is the model capable of dealing with actions related to more than two objects.
>
> **R1**: Please see **Figure 9** of our updated version, where we synthesize a more complex action for our pipeline to learn and obtain visually satisfactory results. As long as the actions can be rendered by the graphics engine, our method could learn to generate them.
>
> ****
>
> > **W2** The motivation of IAScore is clear, however, its reliance on detection introduces potential errors. A more detailed analysis of the reliability and limitations of this metric would make it more credible.
>
> **R2**: Based on our input prompt, the complexity of generated image is relatively low (without generating a complex scene or a large number of objects). We are witnessing the rapid advancement of object detection models, and they can handle this scenario without much concern, as shown in **Figure 6** of our updated version. And we will look into the limitations of IAScore and try to improve it in the future.

---

### Author Response · Authors · 2024-11-23
**Rebuttal Summary**

Dear Reviewers, ACs, SACs, and PCs,

Thanks for your insightful feedback on our manuscript: Interactive-Action Image Generation via Synthetic Physical Priors.

We are encouraged that you find our soundness is good (ro2Y, gBhf, Esr5), the presentation is good (ro2Y, gBhf, Esr5), the contribution is good (ro2Y, gBhf, Esr5).

We have **uploaded a new version** of our manuscript containing more experiments **(Figure 1,6,7,9,10,11)**.

We summarize the main actions taken during the rebuttal:
1) We conduct new experiments to show that our method could handle a more complex action.
2) Additional experiments of more ReVersion results.
3) Additional experiments of our method on real robot dataset DROID.
4) Additional experiments of Stable Diffusion 3 and FLUX.
5) More explanation of IAScore.
6) Clarify the motivation of our paper and several confusions or misunderstandings regarding our paper.

Within this short rebuttal period, we did our best to reply to your questions, and we will incorporate your feedbacks in the final version. In the upcoming interaction period, if you have any questions towards our responses, please feel free to let us know.

Sincerely,

Authors

---

### Meta-Review · Area_Chair_1YC4 · 2024-12-23

**Metareview:**

The objective of this work is to improve the interactive actions in image generation using physical priors. A customized synthetic dataset with interactive actions is provided to finetune the pretrained model to improve the spatial and interactive action aware diffusion models. To account for the domain gap between synthetic and real images, they employ a distillation learning approach for fine-tuning. Experimental results show that the method can improve the interactions between objects.

**Additional Comments On Reviewer Discussion:**

The main concern raised by the reviewers is that this paper is limited in scope. The authors use a small synthetic dataset of a single object manipulation and show some generations where the interactions improve. However, as some reviewers point out, the latest models FLUX can improve this with proper training data and descriptive captions. Also, this approach is not general purpose. The synthetic dataset can't model complex interactions (requires rendering all open world interactions which is quite challenging). Due to the limited scope of this work, I vote for rejecting this paper.

---

### Decision · Program_Chairs · 2025-01-22

Reject